# Sapphire Photonic Crystal Waveguides with Integrated Bragg Grating Structure

Stefan Kefer [1],* , Gian-Luca Roth [1], Julian Zettl [1], Bernhard Schmauss [2] and Ralf Hellmann [1]

1 Applied Laser and Photonics Group, Aschaffenburg University of Applied Sciences, Wuerzburger Strasse 45, 63743 Aschaffenburg, Germany; gian-luca.roth@th-ab.de (G.-L.R.); julian.zettl@th-ab.de (J.Z.); ralf.hellmann@th-ab.de (R.H.)
2 Institute of Microwaves and Photonics, University of Erlangen-Nuremberg, Cauerstrasse 9, 91058 Erlangen, Germany; bernhard.schmauss@fau.de
* Correspondence: stefan.kefer@th-ab.de; Tel.: +49-6021-4206-847

**Abstract:** This contribution demonstrates photonic crystal waveguides generated within bulk planar sapphire substrates. A femtosecond laser is used to modify the refractive index in a hexagonal pattern around the pristine waveguide core. Near-field measurements reveal single-mode behavior at a wavelength of 1550 nm and the possibility to adapt the mode-field diameter. Based on far-field examinations, the effective refractive index contrast between the pristine waveguide core and depressed cladding is estimated to $3 \cdot 10^{-4}$. Additionally, Bragg gratings are generated within the waveguide core. Due to the inherent birefringence of $Al_2O_3$, the gratings exhibit two distinct wavelengths of main reflection. Each reflection peak exhibits a narrow spectral full width at a half maximum of 130 pm and can be selectively addressed by exciting the birefringent waveguide with appropriately polarized light. Furthermore, a waveguide attenuation of 1 dB cm$^{-1}$ is determined.

**Keywords:** integrated photonics; sapphire; Bragg grating; femtosecond laser; photonic crystal waveguide; depressed cladding

## 1. Introduction

Integrated photonic structures generated via femtosecond laser radiation is a rapidly growing area of interest [1]. While the fabrication of several functional structures, such as optical waveguides, couplers, splitters, interferometers and Bragg gratings (BG), within various glass substrates [2–12] is an established process, the scientific community nowadays works on transferring this fabrication method to alternative or novel materials. On the one hand, polymers have been identified as an advantageous platform for integrated photonics [13]. Thus, noticeable progress towards the femtosecond laser-based fabrication of polymer photonic structures was made in recent years [14–19]. On the other hand, especially in the context of developing devices for harsh-environment applications or optical high-temperature sensors, the modification of sapphire substrates by means of femtosecond laser radiation delivers promising results [20]. Most sapphire-based Bragg gratings are generated within highly multimodal air-clad sapphire fibers, either by employing an interferometric setup or the point-by-point writing method. While these devices are capable of handling sensing applications at temperatures up to 1900 °C, they suffer from broad reflection peaks with a spectral full width at half maximum (FWHM) up to 10 nm [21–25]. This circumstance drastically limits their capability of detecting small signal changes and their usage in the outline of multiplexing applications since, in both cases, narrow reflection peaks are essential [26,27]. Currently, few- or single-mode behavior in sapphire fibers, and thus optimal conditions for BG sensing applications, can only be achieved by time-consuming wet-acid etching procedures [28,29] or by a sophisticated melt-in-tube drawing method [30].

There are multiple ways to fabricate optical waveguides in planar, sapphire-based substrates. While processes such as laser ablation, ion beam implantation and reactive ion

etching or wet chemical etching are limited to surface or near-surface structures, generation of waveguides buried deeply in the bulk of these substrates was demonstrated by means of proton implantation and femtosecond laser-based refractive index modification [31,32]. Since buried structures generally promise superior performance due to decreased waveguide asymmetry and, due to the lack of surface interaction, reduced scattering effects, femtosecond laser-based fabrication methods are expected to pave the way towards 3D optical integrated circuits [33]. Bérubé et al. recently demonstrated the first waveguide, buried within a bulk and undoped sapphire substrate, exhibiting single-mode operation at a wavelength of 2850 nm by generating a depressed cladding structure around the waveguide's core via femtosecond laser-based refractive index modification of the material [34].

This study demonstrates an alternative approach to the fabrication of single-mode waveguides at a wavelength of 1550 nm within planar bulk $Al_2O_3$ substrates with a random orientation, based on photonic crystal structures generated via femtosecond laser radiation. Additionally, the photonic waveguide can be equipped with Bragg grating structures, which exhibit outstanding spectral properties for sensing purposes.

## 2. Materials and Methods

Monocrystalline sapphire is one of the most durable, heat- and wear-resistant materials known to mankind. Moreover, its chemical resistance is unmatched by most alternative optical materials [35]. Thus, it is popular in various industrial, medical and scientific application fields [36]. It additionally provides outstanding optical properties in the near-infrared region [37]. Sapphire is built up from rhomboidal crystals, as indicated in Figure 1.

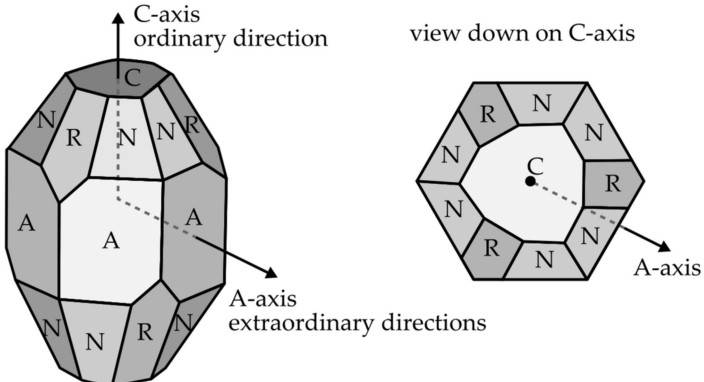

**Figure 1.** Illustration of the sapphire crystal structure. A,C,N and R indicate the respective crystal planes.

From an optical point of view, the crystal's C-axis is of particular interest since it represents the ordinary direction or optical axis of the negative uniaxial crystal. This means that light propagating in parallel to the C-axis of $Al_2O_3$ will not experience any birefringence effects, since both orthogonal polarization components interact with the crystal's ordinary refractive index and are thus equally retarded. However, if light travels in the extraordinary direction, perpendicular to the C-axis, the electrical field components in parallel and orthogonal to the C-axis interact with the crystal's extraordinary and ordinary refractive index, respectively. In this case, the propagating light experiences maximum birefringence [38,39].

While the name photonic crystal fiber or waveguide is mainly correlated to devices exploiting the photonic band gap effect [40,41], this terminology is not solely reserved for suchlike structures. Instead, according to Knight et al., devices based on an effective volume refractive index contrast between a pristine core and a surrounding modification array can be denoted as photonic crystal waveguide (PCWG) as well [42]. All PCWGs manufactured in this study are based on a hexagonal modification matrix around an unmodified core, similar to photonic structures already demonstrated in Nd:YAG crystals [43]. An overview

of the process setup and a cross-sectional schematic of a photonic crystal waveguide are shown in Figure 2.

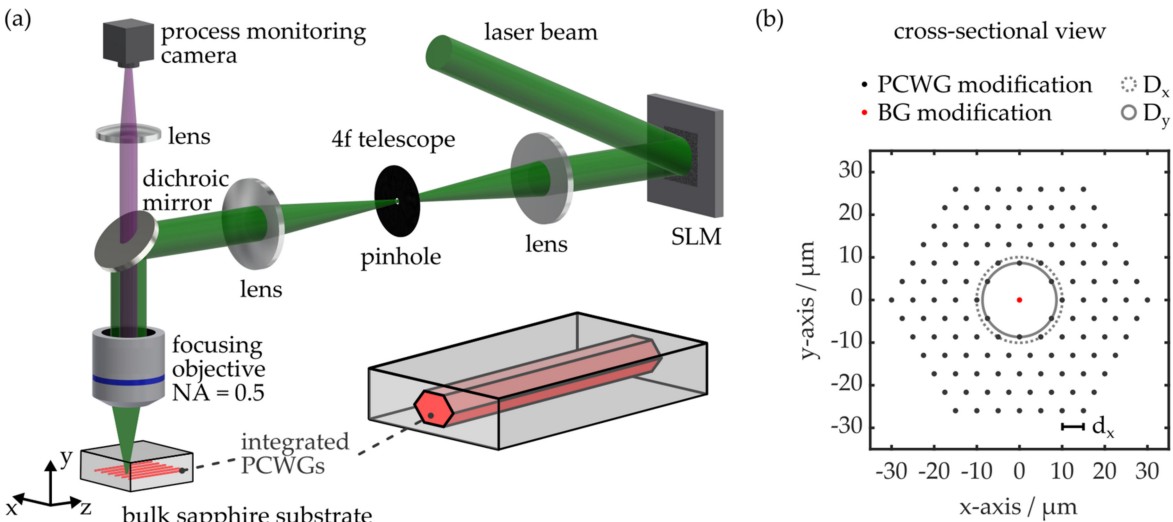

**Figure 2.** (**a**) Schematic of the employed femtosecond laser setup. (**b**) Exemplary cross-sectional schematic of a sapphire photonic crystal waveguide (S-PCWG) with integrated Bragg grating (BG). $d_x$ indicates the distance of two modification lines in x-direction, while $D_x$ and $D_y$ represent the theoretical inner hexagon diameters in x- and y-direction, respectively.

The second harmonic of an ultrashort pulse laser (Pharos-10-600, Light Conversion, Vilnius, Lithuania), with a wavelength of 514 nm, propagates through a beam expansion telescope onto a spatial light modulator (SLM; Pluto VIS 21, Holoeye, Berlin, Germany). The adaptive element, featuring a pixel size of 8 μm with an array size of 1920 × 1080 pixels, is programmed to form a blazed diffraction grating. A 4f telescope (f = 300 mm) images the SLM pattern on the focusing objective, whereas a pinhole within the telescope's focal plane blocks the zeroth order and other extraneous radiation, while the first diffraction order passes the telescope unaffected. It is finally focused with an air objective (EC Epiplan-Neofluar, Carl Zeiss Microscopy, Oberkochen, Germany) to achieve a focal spot diameter $(1/e^2)$ of 1.2 μm. The SLM is additionally used for in situ correction of the wavefront and spherical aberrations, whereas the latter are introduced by the refractive index mismatch between air and sapphire when light is focused into the substrate volume [44,45]. This enables constant focal intensity distributions independent of the writing depth.

The sapphire photonic crystal waveguides (S-PCWGs) are fabricated in a depth of 50 μm within a bulk sapphire substrate ($Al_2O_3$) with an unspecified crystal orientation (AL663025, Goodfellow, Hamburg, Germany). The substrate exhibits a quadratic footprint with an edge length of 25 mm and a thickness of 0.25 mm. Its mean surface roughness $R_S$ is determined as 20 nm by means of white-light interferometry (ContourGT-I, Bruker, Billerica, MA, USA). Every waveguide consists of 120 single modification lines arranged in a lattice-like hexagonal formation, as shown in Figure 2b. All lines are fabricated subsequently in a bottom-to-top fashion using a pulse energy of 54 nJ and a pulse duration of 230 fs, at a repetition rate of 100 kHz. Sample positioning is provided by a high-precision stage (ANT130-XY, Aerotech, Pittsburgh, PA, USA) at a translation speed of 20 mm·s$^{-1}$, which yields a pulse overlap of 1 μm. The generated modification lines are clearly discernible by means of bright-field microscopy, as shown in Figure 3. While they appear black with high contrast, it is noteworthy that the modifications are not hollow. This is confirmed by the fact that liquids do not penetrate the modifications at all when the specimen is immersed. Thus, the resulting lines are solely correlated to refractive index modifications of the bulk sapphire [34].

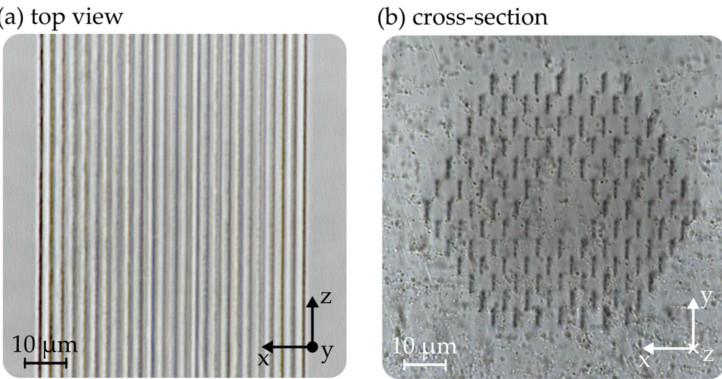

**Figure 3.** Bright-field microscopy images of a sapphire photonic crystal waveguide with 5 μm modification distance: (**a**) top view; (**b**) cross-section.

## 3. Results and Discussion

### 3.1. Photonic Crystal Waveguides in Bulk Planar Sapphire

Multiple waveguides with modification distances ranging from 5 to 9 μm are generated. After polishing the specimen's front faces to a mean surface roughness of 600 nm, light with a wavelength of 1550 nm is coupled into the hexagonal modification arrays by butt coupling a single-mode fiber to each sapphire photonic crystal waveguide's core. Subsequently, the resulting near-field intensity pattern is recorded and the respective mode-field diameter ($1/e^2$ value) is determined [46]. Figure 4 depicts an exemplarily chosen near-field image of an S-PCWG manufactured with a modification distance of 7 μm and the respective intensity distribution in the x- and y-plane of the near-field pattern.

The unambiguous Gaussian shape, observed in both planes, indicates single-mode behavior of the fabricated photonic waveguide. This is also observed for all other waveguide configurations examined in this study. However, for modification distances below 5 μm and above 9 μm, no stable waveguiding is achieved at a wavelength of 1550 nm. Figure 4d shows the resulting mode-field diameters in x- and y-direction of the S-PCWGs with respect to their modification distance $d_x$. It is found that the evolution of the mode-field diameter correlates well to the respective core diameter.

Furthermore, the root cause for the observed mode-field astigmatism is correlated to the physical extension of the photonic structure in the respective plane. While the hexagonal structure exhibits its maximum diameter in x-direction $D_x$, its minimum diameter $D_y$ is found in the y-direction (see Figure 2b). Both parameters are defined by the modification distance $d_x$ according to

$$D_x = 4 \cdot d_x \tag{1}$$

and

$$D_y = 2 \cdot \sqrt{3} \cdot d_x. \tag{2}$$

Please note that the ellipticity of the modification line cross-sections possibly influences the waveguide's astigmatic properties as well (see Figure 3b). However, to the author's best knowledge, this is the first successful demonstration of a photonic waveguide, integrated into bulk $Al_2O_3$, exhibiting single-mode operation within the C-band.

In addition, the numerical aperture of an S-PCWG with a modification distance of 6 μm is determined by acquiring multiple far-field images at increasing distances from the waveguide exit. At every distance, the diameter is quantified by determining the $1/e^2$ value of the respective far-field pattern. The results of the far-field measurement are depicted in Figure 5.

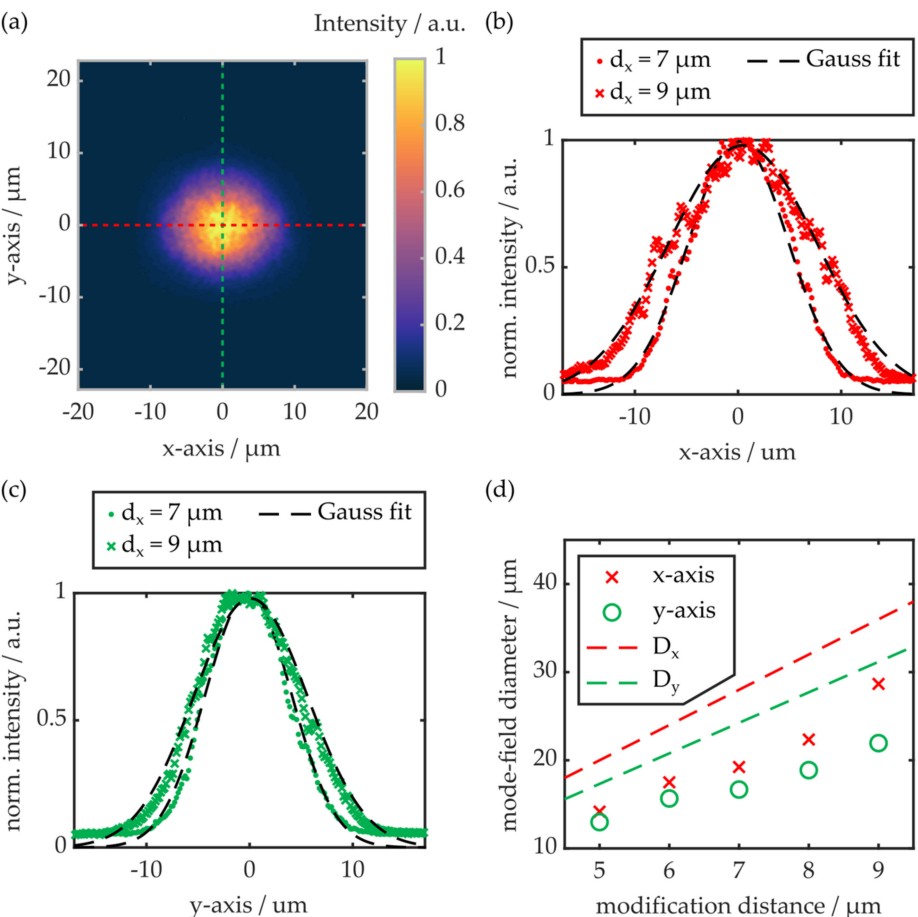

**Figure 4.** (**a**) Exemplary near-field image of an S-PCWG with a modification distance of 7 μm. (**b**) Intensity distribution along the x-axis cross-section for waveguides with modification distances of 7 and 9 μm. (**c**) Intensity distribution along the y-axis cross-section. (**d**) Mode-field diameter as a function of the modification distance. The respective core diameters in x- and y-direction are also indicated.

Again, noticeable astigmatism is observed. The determined numerical aperture, however, is equal on the x- as well as on the y-axis. Based on the waveguide's numerical aperture, NA and the refractive index n of unmodified $Al_2O_3$, which, at a wavelength of 1550 nm, is given as 1.7462 or 1.7384 for electric field components orthogonal or in parallel to the crystal's C-axis, respectively [47], the residual effective refractive index contrast $\Delta n$ of the waveguide core and depressed cladding structure can be estimated according to the relation

$$\Delta n = \sqrt{NA^2 + n^2} - n. \tag{3}$$

Neglecting birefringence, whose impact on the result is less than 0.5%, the residual effective refractive index contrast $\Delta n$ of the waveguide core and depressed cladding is thus estimated as $3 \cdot 10^{-4}$ for the examined waveguide.

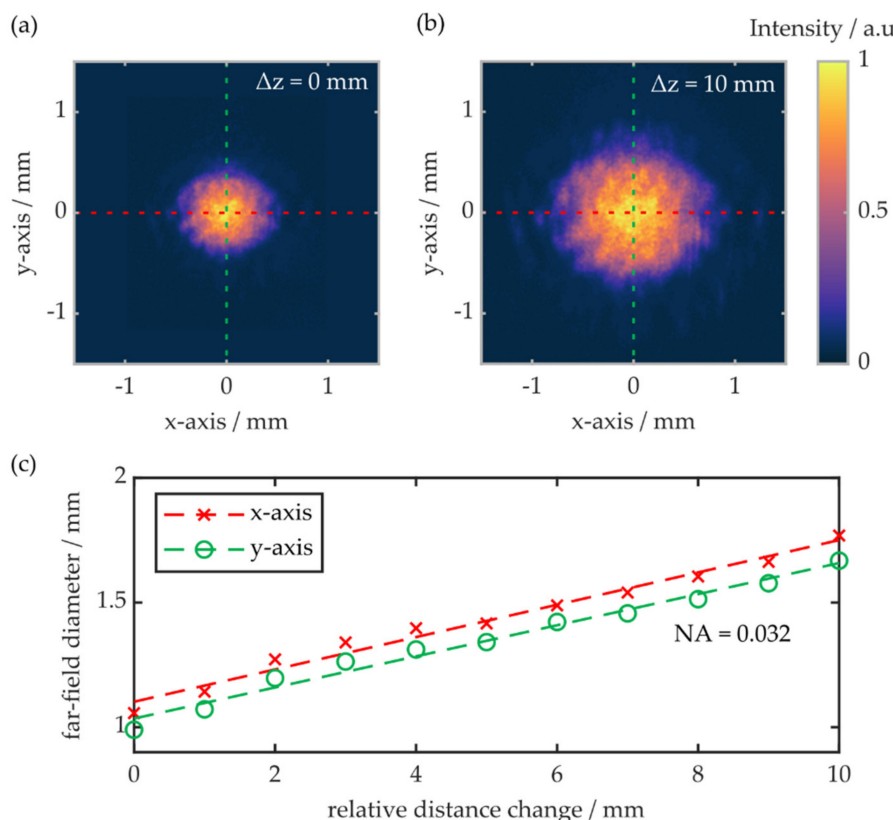

**Figure 5.** (**a**) Far-field image at a relative distance change Δz of 0 mm. (**b**) Far-field image at a relative dis-tance change Δz of 10 mm. (**c**) Respective far-field diameters in x- and y-direction as a function of the relative distance change.

### 3.2. Integrated Bragg Gratings

The demonstrated S-PCWGs can additionally be equipped with a Bragg grating structure, i.e., a periodic refractive index modulation along the waveguide axis. Therefore, an additional modification line is fabricated in the center of the depressed waveguide cladding (see Figure 2b) by employing the point-by-point writing method [26]. For this modification with a length of 5 mm, the pulse energy is kept constant, while laser repetition rate and the translation speed of the substrate are reduced to 500 Hz and 0.44 mm·s⁻¹, respectively. This results in a periodic refractive index modulation with a modification distance, and consequently a Bragg grating period, of 880 nm. The overall manufacturing time of an S-PCWG with integrated BG accumulates to 200 s. Observation of the resulting BG reflection signal is done by means of a commercial-grade interrogation unit (Hyperion si155, Micron Optics), which generates a broadband input signal and evaluates the specific spectral reflection of the photonic structure. The interrogation setup is also equipped with an external polarization control unit. A schematic of the employed evaluation setup as well as the residual Bragg reflection signals for different polarization states are shown in Figure 6.

If the S-PCWG is excited with unpolarized light, the reflection signal exhibits two distinct Bragg peaks with a main wavelength of reflection, or Bragg wavelength $\lambda_B$, of 1529.9 nm and 1536.9 nm, respectively. According to the Bragg relation for a second-order Bragg grating

$$\lambda_B = n_{eff}\Lambda, \tag{4}$$

where $\Lambda$ represents the Bragg grating period and $n_{eff}$ the effective refractive index of the fundamental mode propagating through the S-PCWG, an effective refractive index of 1.7386 and 1.7468 is determined for the respective Bragg peaks. Both results correspond well with the extraordinary and ordinary refractive indices of $Al_2O_3$ crystals around wavelengths of

1530 nm [47]. Thus, the S-PCWG in this specimen exhibits maximum birefringence and, consequently, the waveguide is oriented in the extraordinary direction (see Figure 1). In order to achieve a Bragg signal with only one reflection peak, it is either possible to align the S-PCWG alongside the C-axis of the $Al_2O_3$ crystal structure, which leads to similar refractive indices for all electric field components, or selective excitation with linearly polarized light.

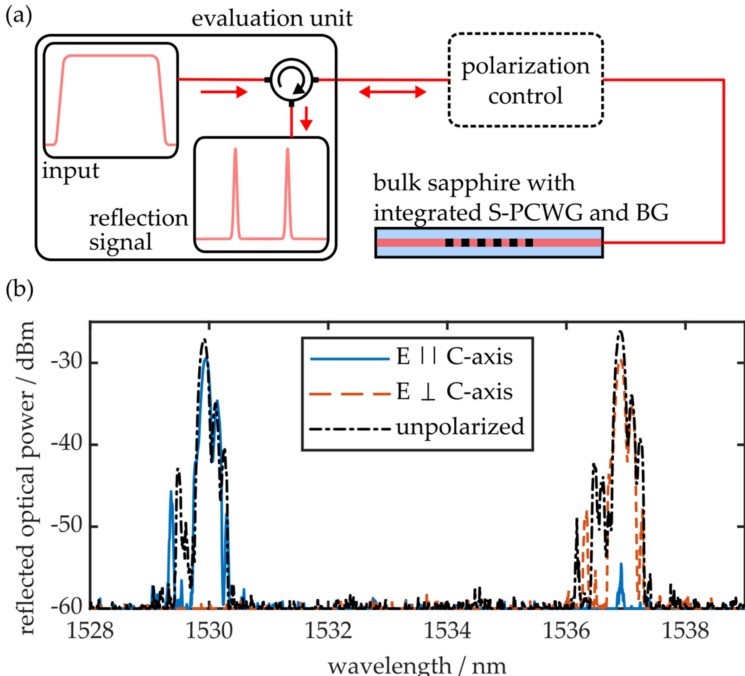

**Figure 6.** (**a**) Schematic of the employed evaluation setup for S-PCWGs with integrated Bragg grating structures. (**b**) Exemplary Bragg reflection signal when an S-PCWG with a modification distance of 5 μm is excited with linearly polarized light, oriented in parallel or perpendicular to the sapphire crystal's C-axis, as well as unpolarized radiation.

In this case, aligning the electric field E in parallel or perpendicular to the crystal's C-axis leads to a single Bragg peak correlating to the extraordinary or ordinary refractive index of the structure (see Figure 6b). Both peaks, however, exhibit a narrow spectral FWHM property of 130 pm. This additionally underlines the single-mode behavior of the demonstrated S-PCWG.

### 3.3. Waveguide Attenuation

The waveguiding losses of an S-PCWG are estimated by employing a Bragg-grating-based, non-destructive method developed by Rogers et al. [48]. For that, an S-PCWG with a modification distance of 5 μm, comprising multiple Bragg gratings along its axis, is fabricated in a bulk sapphire specimen exhibiting a length of 40 mm. All gratings have a length of 5 mm, while each refractive index modulation is manufactured with a distinct grating period of 870, 880, 890 and 900 nm, respectively. A schematic of the Bragg grating positioning within the sapphire specimen is shown in Figure 7a. Depolarized light is successively coupled into the waveguide from both sides, i.e., in forward and backward directions, and, as depicted in Figure 7b, both reflection spectra are recorded. Each grating generates a distinct pair of reflection peaks according to its respective grating period. Since all grating positions are well defined, it is possible to estimate the waveguide's propagation losses by evaluating the ratio of each peak's maximum reflection signal in the forward

direction $R_{fw}$ to its maximum reflection signal in the backward direction $R_{bw}$, as a function of the relative grating position. According to

$$\alpha = \frac{|m|}{4},$$ (5)

the waveguide's attenuation $\alpha$ can then be found by analyzing the absolute gradient of the plot by means of linear regression [49], as shown in Figure 7c. This way, propagation losses of approximately 1 dB cm$^{-1}$, at wavelengths around 1550 nm, are determined. This value is comparable to similar waveguide designs in other materials [18,43].

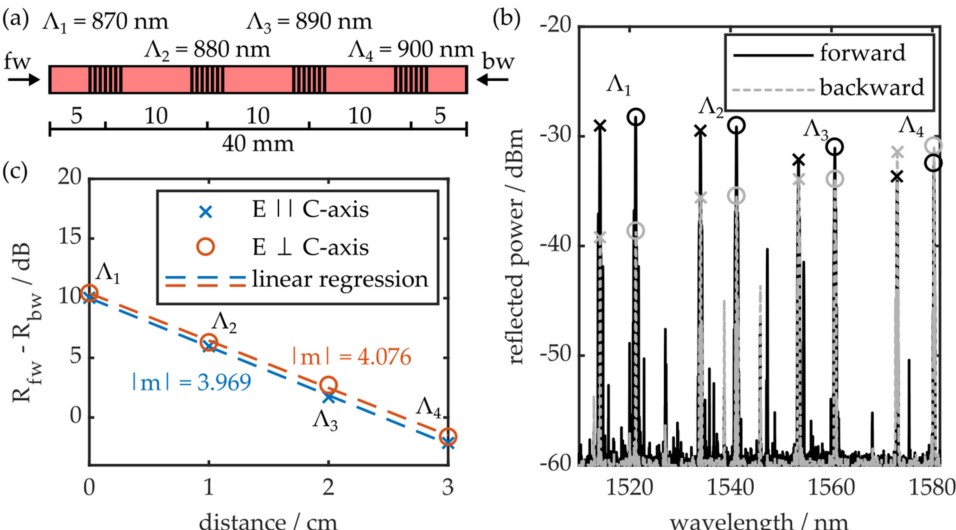

**Figure 7.** (**a**) Schematic of Bragg grating positions within the S-PCWG. (**b**) Bragg reflection signal of both coupling directions. (**c**) Forward and backward coupling peak power ratio as a function of Bragg grating distance.

While a polarization-related deviation of about 2.5% is observed, this is attributed to measurement uncertainties. This experiment additionally highlights the multiplexing capabilities of the demonstrated sapphire-based photonic devices.

## 4. Conclusions

In conclusion, this contribution demonstrates sapphire-based photonic crystal waveguides fabricated within planar bulk Al$_2$O$_3$. The waveguides are fabricated by generating an array of negative refractive index modifications with a hexagonal cross-section around an unmodified core by means of the femtosecond laser-based irradiation of sapphire substrates with a random crystal orientation. Near-field images of the intensity distribution at the S-PCWG exit reveal single-mode behavior, whereas the mode-field diameter can be varied by adapting the modification distance of the waveguide's depressed cladding. Thus, S-PCWGs can be appropriately tailored towards specific applications by straightforward adaptation of the manufacturing parameters. For a wavelength of 1550 nm, single-mode behavior is demonstrated for modification distances from 5 to 9 µm. Far-field measurements of an S-PCWG, fabricated with a modification distance of 6 µm, show a numerical aperture of 0.032, which corresponds to an effective refractive index contrast of approximately $3 \cdot 10^{-4}$ between the core and depressed cladding. Furthermore, Bragg grating structures can be integrated into the S-PCWGs. Two distinct Bragg reflection peaks, caused by the pronounced birefringent properties of Al$_2$O$_3$, are observed. However, a single peak can be selected by exciting the photonic structure with appropriately polarized light. The demonstrated Bragg peaks are highly suitable for measurement or multiplexing applications due to their narrow FWHM of 130 pm. Furthermore, the propagation losses of the demonstrated S-PCWGs are quantified as 1 dB cm$^{-1}$. Conclusively, S-PCWGs with integrated BGs in

planar sapphire substrates pave the way towards integrated photonics in the outline of novel harsh-environment sensing or filtering applications.

**Author Contributions:** Conceptualization, S.K., G.-L.R. and R.H.; methodology, S.K. and G.-L.R.; software, S.K. and G.-L.R.; validation, S.K., G.-L.R. and J.Z.; formal analysis, S.K. and G.-L.R.; investigation, S.K., G.-L.R. and J.Z.; resources, R.H.; data curation, S.K.; writing—original draft preparation, S.K.; writing—review and editing, S.K., G.-L.R., J.Z., B.S. and R.H.; visualization, S.K.; supervision, B.S. and R.H.; project administration, R.H.; funding acquisition, S.K. and R.H. All authors have read and agreed to the published version of the manuscript.

**Funding:** This research as well as the APC were funded by the Bavarian Research Foundation, grant number SK-1388-19.

**Institutional Review Board Statement:** Not applicable.

**Informed Consent Statement:** Not applicable.

**Data Availability Statement:** Data underlying the results presented in this paper are not publicly available at this time but may be obtained from the authors upon reasonable request.

**Conflicts of Interest:** The authors declare no conflict of interest.

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
