# Peer review of "Sapphire Photonic Crystal Waveguides with Integrated Bragg Grating Structure"

_photonics, doi:10.3390/photonics9040234_

Round 1
Reviewer 1 Report
I have the following suggestions to improve the quality of the paper to be further considered for publication in the Photonics journal.
- I am not sure if figure 1 is really required. Can the author explain the necessity of showing the sapphire crystal structure?
- The author has mentioned in the paper that BG can be used for sensing purposes. However, BG has a wide stopband, how it can be used for sensing? It is well suited for filtering applications. According to my opinion, if the Fabry Perot scheme is utilized then it can be most suited for sensing applications due to a narrow transmission in the stopband. Please comment.
- What is the depth of the waveguide writing from the surface of the sample? Please mention.
- Please mention the writing time of a single waveguide structure of a specific length. This information will be useful for the readers.
- Instead of figure 2b, I suggest the author replace a 3D image which will give a better visual understanding of the proposed design.
- Please mention in the introduction section, why the laser writing method is more effective than the conventional waveguide fabrication methods such as litho+RIE? In terms of cost, time, and quality of the waveguides.
Author Response
Dear Reviewer.
Thank you for your feedback and collaboration. Please refer to the attached PDF for detailed responses to your comments.
Kind regards
Stefan Kefer

Reviewer 2 Report
The manuscript presents an alternative approach to fabricate the single-mode waveguides at the third wavelength window within planar bulk Al2O3 substrates with random orientation, based on PhC structures generated by using fs laser radiation. It is an interesting topic for research, however,the following questions should be answered and addressed:
- Propagation loss is one of the most important parameters in photonic crystal fibers. It is recommended to consider the proposed structure from this point of view.
- Higher order modes are the main source of propagation loss. How the destructive effects of high-order modes on the performance of single-mode operation of waveguide are considered?
- Changes in the modification distance (here, dx), leads to changes in the effective refractive index of the photonic crystal structure. How will these changes affect the intensity distribution?
- It is recommended to refer to more references to prove the strengths of monocrystal sapphire.
- It would be better to first determine for what purpose this waveguide is designed. If a specific application is desired, the appropriate structural parameters can be selected.
Author Response

(The authors gave the same response as above.)

Reviewer 3 Report
Dear Editor, thank you for the trust and the opportunity to provide a review of the paper "Sapphire Photonic Crystal Waveguides with Integrated Bragg Grating Structure" Stefan Kefer, Gian-Luca Roth, Julian Zettl, Bernhard Schmauss and Ralf Hellmann.
The paper report about photonic crystal waveguides generated within bulk planar sapphire substrates. The refractive index in a hexagonal pattern around the pristine waveguide core has been modified by a femtosecond lazer. The effective refractive index contrast between pristine waveguide core and depressed cladding is has been determined. A Bragg grating within the waveguide core has been demonstrated. It was shown what each reflection peak exhibits a narrow spectral full width at half maximum of 130 pm and can be selectively addressed by exciting the birefringent waveguide with appropriately polarized light.
The manuscript is well structured, nicely written and I consider it is suitable for the publication in present form. I have only one comment. It is necessary to correct some inaccuracy in line 135. The Figure 4 (c) should be replaced by Figure 4 (d).
Author Response

(The authors gave the same response as above.)

Round 2
Reviewer 1 Report
I am willing to accept the paper in its current form.
Reviewer 2 Report
The revised manuscript can be accepted for publication in Photonics.